# Urinary Biomarkers for Diagnosis and Prediction of Acute Kidney Allograft Rejection: A Systematic Review

**DOI:** 10.3390/ijms21186889

**Published:** 2020-09-19

**Authors:** Francesco Guzzi, Luigi Cirillo, Elisa Buti, Francesca Becherucci, Carmela Errichiello, Rosa Maria Roperto, James P. Hunter, Paola Romagnani

**Affiliations:** 1Department of Experimental and Clinical Biomedical Sciences “Mario Serio”, University of Florence, 50134 Florence, Italy; luigi.cirillo@unifi.it (L.C.); paola.romagnani@unifi.it (P.R.); 2Nephrology and Dialysis Unit, Meyer Children’s University Hospital, 50139 Florence, Italy; elisa.buti@meyer.it (E.B.); francesca.becherucci@meyer.it (F.B.); carmela.errichiello@meyer.it (C.E.); rosa.roperto@meyer.it (R.M.R.); 3Nuffield Department of Surgical Sciences, Oxford Transplant Centre, Churchill Hospital, University of Oxford, Oxford OX3 7LJ, UK; james.hunter@nds.ox.ac.uk

**Keywords:** kidney transplantation, kidney graft, T-cell-mediated rejection, antibody-mediated rejection, diagnostic test accuracy

## Abstract

Noninvasive tools for diagnosis or prediction of acute kidney allograft rejection have been extensively investigated in recent years. Biochemical and molecular analyses of blood and urine provide a liquid biopsy that could offer new possibilities for rejection prevention, monitoring, and therefore, treatment. Nevertheless, these tools are not yet available for routine use in clinical practice. In this systematic review, MEDLINE was searched for articles assessing urinary biomarkers for diagnosis or prediction of kidney allograft acute rejection published in the last five years (from 1 January 2015 to 31 May 2020). This review follows the Preferred Reporting Items for Systematic Reviews and Meta-analysis (PRISMA) guidelines. Articles providing targeted or unbiased urine sample analysis for the diagnosis or prediction of both acute cellular and antibody-mediated kidney allograft rejection were included, analyzed, and graded for methodological quality with a particular focus on study design and diagnostic test accuracy measures. Urinary C-X-C motif chemokine ligands were the most promising and frequently studied biomarkers. The combination of precise diagnostic reference in training sets with accurate validation in real-life cohorts provided the most relevant results and exciting groundwork for future studies.

## 1. Introduction

The growing call for precision medicine justifies a trend shift towards the implementation of new prognostic and diagnostic biomarkers in many fields of medicine. Progress in molecular and biomarker technology now permits the possibility to tailor and customize clinical and therapeutic approaches to the specific needs of a single patient, for a variety of medical conditions. Kidney diseases are no exception, with biomarkers constantly gaining more ground in the management of acute kidney injury (AKI), glomerulopathies, and chronic kidney disease (CKD) [1,2,3]. Also, in the setting of kidney transplantation, precision medicine is rapidly moving forward, with biomarkers a significant part of this trend. In kidney transplantation, biomarkers have been studied for early recognition and diagnosis of disease recurrence, delayed graft function (DGF), infections, and acute and chronic allograft rejection [4]. Since the 1970s, biomarkers have been studied for organ quality assessment prior to transplantation [5,6] and post-transplant evaluation [7]. However, in clinical practice, few genuine biomarkers have emerged, and clinicians still largely rely on serum creatinine and proteinuria monitoring. Novel biomarkers could be of great help not only for early recognition of allograft disease, but also for monitoring disease activity, optimizing the need for invasive biopsies, predicting the effectiveness and safety of a certain treatment, and tailoring the management of each single patient to their specific needs [4,8].

Despite near-optimal immunosuppressive regimens and accurate therapy compliance, kidney transplants still suffer from potentially preventable acute rejection (AR) episodes. AR early identification is important for preserving nephron mass and aiding long-term allograft survival [9]. The gold standard for AR diagnosis is histological examination of a kidney biopsy. The biopsy can then be interpreted with the help of the Banff classification (created in the 1990s and periodically revised), which describes acute lesions according to two mechanistic pathways: T-cell-mediated rejection (TCMR) and antibody-mediated rejection (ABMR) [10]. However, a biopsy is an invasive procedure, may not be straightforward to perform and can be complicated by major bleeding. In addition, potential sampling errors, inter-observer variability, and elevated costs make allograft biopsy impractical for continuous monitoring of the graft over time. Urine samples, as a readily available and direct product of the allograft, with minimal influence from systemic inflammation, are a more desirable source for AR biomarkers. Very recently, narrative reviews explored the use of biomarkers in the diagnosis of AR [11,12]. However, to our knowledge, the most recent systematic review assessing urinary biomarkers’ ability for allograft AR diagnosis in kidney transplant patients included papers published until 2015 [7]. The most relevant articles and findings in the field before 2015 have been also thoroughly summarized elsewhere [13,14,15]. Up to 2015, no urinary biomarker was validated in sufficiently robust trials to be translated into clinical practice independent of traditional surveillance and diagnostic methods.

The aim of this systematic review is to perform a methodical analysis and to summarize important results coming from the most recent literature (2015–present) evaluating urinary biomarkers and their performance as diagnostic and/or predictive tools for kidney allograft acute rejection. 

## 2. Results

### 2.1. Included Studies

The original literature search yielded a total of 314 citations. Of these, 251 studies were discarded after evaluation of title and abstract in the eligibility process. The remaining 63 studies were reviewed full-text for inclusion in the study. Twenty-five studies were excluded, as detailed in Figure 1.

The main reason for exclusion was the evaluation of a different outcome instead of AR (e.g., chronic or late rejection, graft dysfunction, graft failure). Experimental studies, one trial protocol, and one letter were also excluded. A total of 38 remaining articles, published between 1 January 2015 and 31 May 2020, were finally included. No additional articles were included from the reference lists.

### 2.2. Study Characteristics

Table 1 summarizes the most relevant characteristics of each of the 38 included studies. Twenty-nine were single center studies, while nine were multicenter collaborations. Among the 32 studies assessing urinary biomarkers in the diagnosis of acute rejection, the majority of them, 18/32, were designed as case-control, while 14/32 as cross-sectional studies. Among the six studies assessing the ability of urinary biomarkers to predict acute rejection at variable time-points before the episode, five studies analyzed prospectively collected data and one study analyzed retrospectively collected data. Population characteristics, patient selection protocols, inclusion and exclusion criteria, as well as sample size, were heterogeneous between studies. Sample size varied from 15 to 396 kidney transplant patients, and occasionally more than one urine sample was obtained from the same patient. Although all of the included studies were published after January 2015, some of them enrolled patients transplanted from the early 2000′s. Table 1 also highlights the year of study population enrolment to help the identification of possibly overlapping populations and to assess the appropriateness of the Banff classification used in each study. The majority of the included studies applied up to date Banff classification, with ten studies apparently using the 1997 version or not reporting the year. Using an outdated classification could be a source of bias mostly for studies assessing ABMR, as discussed later. For one study, the application of the Banff classification was not clearly stated. Studies were also heterogeneous in terms of the considered outcome. Among the 38 included studies, 15 specifically addressed TCMR only, whereas just two exclusively focused on ABMR. Sixteen studies assessed the combination of TCMR and ABMR, while five studies did not specify the characteristics of the observed acute rejection (Table 1).

### 2.3. Biomarkers

The included studies were split into diagnostic and prediction studies. To be considered a diagnostic study, the collection of a urine sample should be performed on the day that AR was suspected or when per-protocol biopsies were planned. There were a few exceptions to this rule where sample collection occasionally occurred up to seven days before biopsy. For prediction studies, urine samples were collected at any time point post-transplant and, in this analysis, these ranged from day one up to six months post-transplantation. Among the various techniques for targeted analysis of known urinary biomarkers, ELISA and RT-PCR were the most frequently utilized. Mass spectrometry, nuclear magnetic resonance spectroscopy, liquid chromatography, RNA expression, and transcriptome analysis by RNA-Seq were employed for unbiased metabolomics, proteomics, and genomic profiling and for detection and identification of urinary exosome proteins. All biomarkers are detailed per category in Table 2.

In accordance with previous studies, the most extensively assessed urinary biomarkers were C-X-C motif chemokine ligand 9 (CXCL9) and 10 (CXCL10), usually adjusted for urinary creatinine concentration. In detail, 12/38 (32%) studies either addressed CXCL9 and CXCL10 alone, in combination, or in the context of particular scores or formulas [16,17,24,26,34,35,36,44,47,50,51,52]. Other directly targeted cytokines and interleukins were chemokine ligand 2 (CCL2), also known as monocyte chemoattractant protein 1 [22,31,43], CXCL13 [29], interleukin 10 (IL10) with interferon gamma (IFNγ) [46] and tumor necrosis factor alpha (TNFα) [32].

Unbiased metabolomic analysis and untargeted profiling revealed multiple urinary metabolites as potential biomarkers of AR: nicotinamide adenine dinucleotide (NAD), nicotinamide adenine dinucleotide phosphate (NADP), nicotinic acid, 1-methylnicotinamide (MNA), gamma-aminobutyric acid (GABA), cholesterol sulfate, homocysteine [18], the combination of alanine, citrate, lactate, and urea [21,27], and the combination of guanidoacetic acid, methylimidazoleacetic acid, dopamine, 4-guanidinobutyric acid, and l-tryptophan [25].

Urinary proteins of interest were neutrophil gelatinase-associated lipocalin (NGAL) [22,40], liver-type fatty acid-binding protein (LFABP) [22], human epididymis protein 4 (HE4) [22], matrix metalloproteinase 7 (MMP7) [47], soluble T cell immunoglobulin mucin domain 3 (sTIM3) [37]. The presence and diagnostic performance of urinary extracellular vesicle (exosome) proteins, derived from inflammatory cells and collected with the help of nano-membrane or immune-magnetic capture, was investigated by three studies [28,32,52].

Direct RT-PCR was used to identify targeted RNAs like programmed cell death protein1 (PD1) mRNA, forkhead box P3 (FOXP3) mRNA, and micro (mi) RNAs (miR-142-3p, miR-155-5p) [30,34,38,48], while genome-wide or transcriptome analysis were applied for the unbiased identification of circular and long noncoding RNAs [23,49].

The amount of urinary CD4+ and CD8+ T cells, tubular epithelial cells (TEC), podocalyxin (PDX)-positive, CD10+ or epithelial cell adhesion molecule (EPCAM)-positive cells was determined by flow cytometry and compared with biopsy results [20].

Two studies [35,44] investigated the diagnostic accuracy of the *CTOT4 formula* (CXCL10 mRNA, CD3ε mRNA, 18S rRNA), previously validated by the CTOT-4 (clinical trials in organ transplantation-4) multicenter study group [54]. CXCL10 was also included in newly derived scores such as the *Q score*, composed of six DNA, protein, and metabolite urinary biomarkers (cell-free DNA, methylated cell-free DNA, clusterin, total protein, creatinine, and CXCL10) [17], while CXCL9 and CXCL10 genes were included in the *uCRM score* [24]. Finally, the *ABMR score* comprised more than 130 unique metabolites [41].

### 2.4. Quality Assessment

Studies reporting diagnostic test accuracy (DTA) analysis (29/38) were graded for risk of bias and applicability concerns according to the QUADAS-2 tool (Table 3). Risk of bias was frequently high for patient selection (14/29, 48%) and index test (23/29, 79%).

The most frequent reasons for high risk of bias were the selection of the study population by case-control design, which was the case for the majority of the studies, the exclusion of the typical confounding of a real-life setting, the absence of threshold definition and independent validation. For example, when control patients were selected among stable patients without performing allograft biopsy, or only among normal histology patients, and the obtained thresholds were not tested in a randomly selected validation group, the study was highlighted for high risk of bias in patient selection and index test (Table 3). This then raised the possibility of an increased risk of over-fitting association and unrealistic DTA performance and, therefore, concerns for applicability. The ideal control patients were randomly (or in a cross-sectional fashion) selected, all having had an allograft biopsy (per indication or per protocol) with various histological diagnosis (e.g., normal histology; acute tubular necrosis, ATN; interstitial fibrosis and tubular atrophy, IFTA; chronic allograft nephropathy, CAN; BK virus nephropathy, BKVN; recurrence of the primary disease on the allograft). Only 5/29 studies were found to have a low risk of bias in both patient selection and index test. Allograft histology, according to Banff classification, was the reference standard for AR diagnosis, with histology grading usually assigned in a blinded fashion with respect to the index test results. Since urinary samples were frequently obtained for all included patients, prior to a diagnostic allograft biopsy, and all included patients were evaluated in the DTA analysis, a low risk of bias was frequently identified in the flow and timing domain. The QUADAS-2 tool does not include publication bias (PB) as one of the variables and, in the context of this review, it is difficult to formally assess PB. Given the broad variety of different biomarkers that were assessed and the absence of a meta-analysis, performing formal PB assessment such as Egger’s test, Deek’s test or the construction of a funnel plot was not possible. It is also recognized that the assessment of PB in data synthesis of DTA data is challenging with limited reliability [55].

### 2.5. Summary of the Results

Table 4, Table 5 and Table 6 provide a detailed summary of each study results. When DTA analysis was available (29 studies), the results are summarized in Table 4 for diagnostic studies (24/38) and Table 5 for prediction studies (5/38). Descriptive results from the remaining nine studies are briefly reported in Table 6. For each DTA study, the particular outcome of interest and characteristics of the control population are reported with sample size for each group included in the final DTA analysis. The urinary biomarker of interest, thresholds (when available) and test design (training, validation, or particular comparisons between groups) are also detailed. For prediction studies, time from transplantation to urinary biomarker analysis is also reported (between 1 day to 6 months post-transplantation). Sensitivity, specificity, positive and negative predictive values (PPV, NPV), and area under the receiver operating characteristic curve (AUC) are reported as measures of diagnostic test accuracy when available (Table 4 and Table 5) and results are in bold text when arising from validation cohorts. Results confirmation in at least one validation cohort was available in less than one third of studies (7/29, 24%). Of these, two were case-control studies [25,33], while the others were the previously mentioned five cross-sectional studies with the lowest risk of bias score [16,17,21,41,45]. Sensitivity and specificity values were highly variable between studies, ranging from 9% to 100% and from 34% and 100%, respectively. PPV and NPV were also variable, ranging from 15% to 98% and from 32% to 100%, respectively.

#### 2.5.1. Acute Rejection Diagnosis

Among studies with the lowest risk of bias, only three studies [16,17,45] yielded a very good (0.8–0.9) or excellent (> 0.9) performance as diagnostic AUC (Table 4). All of these studies provided diagnostic accuracy measure for the diagnosis of AR, considering both TCMR and ABMR as outcome of interest. Tinel et al. found that the combination of urinary CXCL9 and CXCL10 could distinguish AR patients among almost three-hundred heterogeneous patients with an AUC of 0.70 [16]. These results strengthened the good performance previously described, among dysfunctional allografts, separately for CXCL9 (AUC 0.71) and CXCL10 (AUC 0.74) by Rabant and colleagues [51]. Yang et al. separately validated the so-called *Q score* in two validation cohorts for the diagnosis of AR. A *Q score* ≥ 32 maintained an excellent diagnostic performance (AUC 0.96) also when validated in the entire study population (*n* = 364), with high PPV and NPV (87–98%) [17]. Banas et al., after identifying a urinary metabolite signature with good diagnostic performance for TCMR [27], validated it in a cohort of 109 patients for the diagnosis of AR with and AUC of 0.71 [21]. Through unbiased metabolomics, Sigdel et al. identified a signature of eleven urinary peptides able to segregate AR patients from normal histology, chronic allograft nephropathy and BK virus nephropathy patients with an excellent AUC of 0.94 in validation cohort [45]. The same authors proved a urine cell sediment gene expression-based score (*uCRM score*) able to diagnose AR with 96.6% accuracy and potentially quantify the degree of injury [24].

#### 2.5.2. T-Cell-Mediated Rejection Diagnosis

The previously mentioned study by Tinel et al. also provided separate outcome analysis for CXCL9 and CXCL10, with the best performance for TCMR diagnosis with a NPV of 98% and a very good AUC of 0.81 [16]. Also of note, CCL2, at a threshold level of 198 pg/mL, yielded very good performance (AUC 0.81) for TCMR identification among a population of 300 normal and dysfunctional grafts in the study by Raza et al. [43]. Urinary exosome proteins were investigated in two case-control studies for the diagnosis of TCMR. Lim et al. found significantly higher urinary tetraspanin-1 (TSPAN1) and hemopexin (HPX) expression levels in TCMR patients with good diagnostic performance (AUC 0.74) [28], while Park et al. reported the initial results of an optimized integrated kidney exosome analysis (iKEA) able to distinguish TCMR from normal histology patients, with a very good performance (AUC 0.84) in a small validation cohort [33].

#### 2.5.3. Antibody-Mediated Rejection Diagnosis

The study from Blydt-Hansen et al. was the only one to specifically evaluate the diagnostic performance for ABMR diagnosis [41]. The authors tested and validated the use of the *ABMR score,* with a good sensitivity (78%) and specificity (83%), NPV of 96%, a good performance (AUC 0.76 in validation), and the ability to provide a stratification from negative—indeterminate—to positive ABMR patients [41].

#### 2.5.4. Acute Rejection, TCMR, and ABMR Prediction

Among prediction studies (Table 5), high risk of bias was often identified for patient selection and index test. However, good performances for AR prediction were obtained by three months post-transplant for CXCL9 and CXCL10 levels [26], and seven days and one month post-transplant for TNF-alpha levels [32]. The well-conducted study by Rabant et al. found both urinary CXCL9 and CXCL10, adjusted for urinary creatinine concentration, to have high NPV (89 to 93%) for AR at one and three months post-transplantation. CXCL10 yielded the best predictive performance (AUC 0.72) at one month post-transplantation, at the threshold of 2.79 ng/mmoL [50]. For TCMR prediction, post-transplant CXCL10 and miR-155-5p levels yielded positive results [34], while for ABMR prediction six months albuminuria was investigated [42].

## 3. Discussion

With this systematic review, we critically summarize the results of the last five years research, the latest advances, and highlight the most frequent limitations of studies assessing urinary biomarkers for the diagnosis or prediction of acute allograft rejection. We focused on study design, distinction between TCMR and ABMR setting, evaluation of confounding (e.g., DGF, infections, calcineurin inhibitors nephrotoxicity), comparison with the gold standard of diagnosis (both for cases and controls), and presence of estimates of the biomarker(s) performance in validation.

The main finding was the strengthening in evidence for the clinical utility of urinary C-X-C motif chemokine ligands (in particular for the diagnosis of TCMR) alone or in combination with other biomarkers as in the *Q score* (cell-free DNA, methylated cell-free DNA, clusterin, total protein, creatinine, and CXCL10) or in the CTOT-4 formula. CXCL9 and CXCL10 had AUC ranging from 0.67–0.88 with a NPV ranging from 84–98% for AR diagnosis and AUC ranging from 0.50–0.97 with a NPV ranging from 71–96% for AR prediction. Signatures of urinary peptides and metabolites identified through unbiased proteomic and metabolomics, and a cluster of urinary cell pellet genes (*uCRM score*) were also established for the diagnosis of AR, net of some limitations for their introduction in clinical practice. Confounding outcomes need always to be considered due to potential overlap in diagnosis. For example, urinary chemokines are also elevated in allograft BK virus nephropathy (as discussed below), urinary NGAL was proposed as early predictor of DGF [56], and as a biomarker of CNI toxicity [57], while urinary miRNAs dysregulation has been linked to interstitial inflammation and tubular atrophy [58]. For the first time Tinel and colleagues demonstrated that considering (instead of excluding) potential confounding factors (i.e., urinary tract infection and BK virus reactivation) in a diagnostic multi-parametric model could optimize its performance [16]. A model combining eight parameters (recipient age, sex, eGFR, DSA presence, signs of urinary tract infection, BKV blood viral load, CXCL9, and CXCL10) could reach AR diagnosis with high accuracy (AUC: 0.85, 0.80–0.89), paving the way for new studies combining urinary biomarkers with clinical characteristics to reach the highest clinical relevance and provide targeted therapy for our patients.

Up to 2015, almost ninety non-redundant molecules were identified as urinary biomarkers of AR, participating in different pathways such as complement activation, antigen presentation, and inflammation signaling [15]. Urine was the most frequent matrix of choice for these analyses, and studies were often limited by small sample size and case-control design, no histology in the control cohorts, lack of confounding adjustment, lack of a validation set, and technical difficulties with procedure standardization and costs [15]. Although serum creatinine levels and proteinuria monitoring are well established biomarkers used by transplant physicians to suspect AR, they lack both sensitivity and specificity, and they are of little help in the prediction phase, in detecting subclinical rejection, and in differential diagnosis between AR, infections, drug toxicity, and acute tubular necrosis [14,59]. In a study of 281 consecutive biopsies, indicated by an increase in serum creatinine levels, only 27.8% revealed any sign of AR [51]. Conversely, subclinical rejection (i.e., rejection without clinical dysfunction) was found in over 40% of patients with normal renal function in the presence of anti-HLA de novo donor-specific antibodies (DSA) [60]. Proteinuria is common after kidney transplant and, although widely used as a biomarker of renal disease and despite its value as an independent predictor of long-term graft survival, it could also be sign of post-transplant primary disease recurrence (e.g., focal-segmental glomerulosclerosis), infections (e.g., CMV), immunosuppressive medication toxicity, or systemic (e.g., new-onset diabetes) and urologic complications (e.g., ureteral stenosis) [59,61]. DSA monitoring is currently considered the primary biomarker for ABMR but, despite the increasing ability to detect low level of DSAs, their positive predictive value is low, so that up to 60% of patients showing de novo DSA do not show any sign of AR at biopsy [60].

Continuous advances in molecular techniques and the “-omics” sciences have helped to identify many potential new blood and urine biomarkers for the diagnosis and prediction of kidney allograft AR in the last two decades. Of note, elevated pretransplant serum CXCL9 and CXCL10 levels were found to be associated to increased risk of early and severe AR and graft failure [62,63,64]. Subsequently, among urine-derived proteins, a 2012 study found CXCL9 and CXCL10 to be considerably elevated in patients experiencing either AR (clinical or subclinical) or BK virus infection (86% sensitivity and 80% specificity for CXCL9; 80% sensitivity and 76% specificity for CXCL10), but they were not able to distinguish between the two conditions [65]. These results were reinforced by the 2013 CTOT-1 study, which found that low urine CXCL9 measured at 6 months post-transplant identified a subset of patients at low-risk for AR development (92% NPV for Banff ≥1A TCMR) and predicted allograft stability up to 24 months post-transplant (93-99% NPV) [66]. With the help of mass spectroscopy, elevated beta2-microglobulin levels were identified as strongly correlated with AR (83% sensitivity, 80% specificity, 89% PPV, 71% NPV) and then validated by ELISA in the urine of AR patients [67]. Cytotoxic proteins perforine and granzyme B urine mRNAs were proposed to noninvasively diagnose AR (respectively with 83% sensitivity, 83% specificity, and 79% sensitivity, 77% specificity) [68] and Treg marker FOXP3 was shown to predict reversal of AR (90% sensitivity, 73% specificity) [69]. T-cell immunoglobulin-3 domain, mucin domain mRNA expression (Tim-3, also known as hepatitis A virus cellular receptor 2) in urinary cells was found to be able to discriminate AR from other causes of acute graft dysfunction (calcineurin inhibitor nephrotoxicity or interstitial fibrosis and tubular atrophy) with an AUC of 0.96, 89% PPV and 94% NPV [70]. A 2013 multicenter study from the CTOT-4 study group later identified a 3-gene urinary mRNA signature (CD3ε mRNA, CXCL10 mRNA, 18S rRNA) able to discriminate acute TCMR from no rejection in indication biopsies, with an AUC of 0.74, 79% sensitivity and 78% specificity in a validation set [54]. Also, noncoding miRNAs (e.g., miRNA-10a, miRNA-10b, miRNA-210), although limited by the easy degradation, proved to be detectable in the urine, and in particular low miRNA-210 levels discriminated patients affected by AR from stable control transplant patients (74% sensitivity, 52% specificity) [71].

Our systematic analysis of the more recent literature details the accuracy of a variety of urinary biomarkers for allograft AR with the objective of allowing transplant physicians early diagnosis and prediction of rejection episodes, and differential diagnosis with other causes of allograft dysfunction. A correct histologic diagnosis of AR is essential during the process of new biomarkers validation and the Banff criteria are considered the gold standard for biopsy evaluation. The diagnostic criteria for TCMR have essentially undergone no major change in the last decade with lymphocytic infiltrate of tubules (tubulitis) and larger vessels (vasculitis) being the main descriptive features. The severity of these lesions is graded according to the degree of lymphocytic infiltrate per high-powered field. On the other hand, ABMR criteria has continuously evolved in recent years–thus highlighting the great importance of applying an up to date classification in this setting–with the recognition of its variable histologic presentation [72,73]. Original criteria established in 2000s included active tissue injury, immunohistologic evidence of peritubular capillary complement split-product C4d deposition and circulating DSA. Subsequent studies demonstrating the presence of ABMR also in lacking detectable C4d staining biopsies [74], pushed the Banff Working Group in 2013 to the major change in the ABMR criteria, removing the requirement for C4d detection [75]. The most recent changes in 2017 included removing the requirement for documented circulating DSA in the setting of positive C4d staining and microvascular inflammation and included the use of AMR-associated gene transcripts panels [10].

The ideal biomarker should be readily available, accurate, inexpensive, standardized, repeatable, and noninvasive and would be useful to reduce the need for protocol biopsy and enable early targeted intervention. The chance of finding an ideal biomarker with high sensitivity, specificity, PPV and NPV is small. However, not all biomarkers need to be highly sensitive and highly specific at the same time, depending on the clinical question they are going to answer. Therefore, targeting specific populations and accepting lower predictive values in certain variables may be a better strategy. For example, to confirm the need for allograft biopsy in a population at high risk for AR (thus providing biopsy to the correct patients), a test with high sensitivity, and low false negative rate, would be the most useful. On the contrary, to propose diagnostic biopsies in a population at low risk for AR (thus avoiding unnecessary per-protocol biopsies), a test with high specificity, and low false positive rate, would be the test of choice. Also, TCMR and ABMR are different clinical entities and it is unrealistic, on current evidence, to hope for a biomarker that will accurately predict AR in both forms in a typical population of transplant patients with possible confounding.

Our systematic review has some limitations. The heterogeneity of the included studies did not permit to detail the many facets of individual study results, especially the more complex ones, to stick with the systematic review question. For space restraints, tables only report the major findings of each study, limited to urinary biomarkers. A narrative synthesis of the most promising results was applied to improve readability and a meta-analysis could not be performed. From our work, overall good quality studies emerged, many with DTA analysis and some comprising a thorough validation process yielding a very good to excellent diagnostic performance. Although specific forms of bias were assessed using QUADAS-2 publication bias could not be formally assessed and the authors acknowledge this can overestimate the weight of positive results. Weaknesses of the included studies were often the use of small cohorts obtained by case-control selection yielding inflated predictive values, the exclusion of confounding, unclear or out of date Banff classification application, the absence of validation cohorts, and lack of hypothesis-driven approach. In fact, the biomarker discovery process should not only consist of a training phase (i.e., a case-control study), but also comprise independent validation in a prospective study and confrontation with real-life clinical setting.

## 4. Materials and Methods

### 4.1. Literature Search

This review follows the Preferred Reporting Items for Systematic Reviews and Meta-Analysis (PRISMA) guidelines [76]. The objective of the study, search strategy, inclusion and exclusion criteria, and study evaluation method were planned in advance, refined, and approved by all authors. MEDLINE was searched from 1 January 2015 to 31 May 2020. Key terms like “kidney”, “renal”, “transplant/transplantation”, “urine/urinary”, “marker/biomarker”, and “rejection” were combined in the search strategy. Additional relevant articles were searched from scanning reference lists of included studies and added if not detected by the original literature search.

### 4.2. Selection Process

The first screening by title and abstract was separately performed by two authors (F.G., L.C.) in the eligibility process. Original articles were selected if they assessed one, more, or a combination of urinary biomarker(s) and their performance in diagnosis or prediction of kidney allograft AR. Abstracts, reviews, studies assessing biomarkers from other matrix (e.g., blood samples or histology staining), and studies specifically evaluating different outcomes (e.g., chronic rejection, infection, or allograft survival) were excluded. In the inclusion process, selected articles were then independently full-text reviewed by two authors (F.G., L.C.). Any disagreement between the two investigators was discussed and solved with the help of all authors.

### 4.3. Data Collection and Analysis

Data from each of the included studies were collected with the help of a pre-specified spreadsheet and extraction table refined by all authors. Study design, single or multicenter patient collection, sample size, years of enrollment, urinary biomarker(s) of interest (i.e., index test), the Banff classification used for histological AR diagnosis (i.e., reference standard) and the addressed outcome(s) were collected in a descriptive table. Studies were distinguished between diagnostic and predictive. Diagnostic studies were usually collecting urine samples on the day of the diagnostic biopsy while predictive studies were analyzing urine samples collected before AR development. Studies that reported DTA data, such as sensitivity, specificity, PPV, NPV, and AUC were evaluated for risk of bias and applicability concern using the Quality Assessment Tool for Diagnostic Accuracy Studies-2 (QUADAS-2), a tool for quality evaluation of diagnostic accuracy studies [77]. The most important items for a positive evaluation included; a cross-sectional study design; avoiding patient selection bias and inappropriate exclusion; the definition of the index test (biomarker) threshold in a training set and its validation in a separate set of patients; and compliance with the correct histological definition of AR as a standard reference for all patients included in the analysis. Due to the great heterogeneity of the included studies, a meta-analysis was not performed, and a narrative synthesis of the results was preferred.

## 5. Conclusions

In recent years, numerous studies joined the challenging quest for urinary biomarkers in diagnosis and prediction of acute kidney allograft rejection. Authors must face the difficult task to allow for mediating between the need for a precise setting and reference standard diagnosis (to develop the most precise biomarkers), and the need for their validation in the most heterogeneous population of kidney allograft patients (to increase clinical utility). Urinary chemokines CXCL9 and CXCL10, alone or in combination with others, are the most frequently used and the most promising biomarkers, but multi-parametric clinical and laboratory models could represent the best strategy for future studies. Remarkable advances have been made on the path of allowing a more precise allocation of resources, helping clinicians to move from the standard protocol/indication biopsy dichotomy, to reduce unnecessary immunosuppression, and to improve kidney allograft outcomes in the long-term.

## Figures and Tables

**Figure 1 ijms-21-06889-f001:**
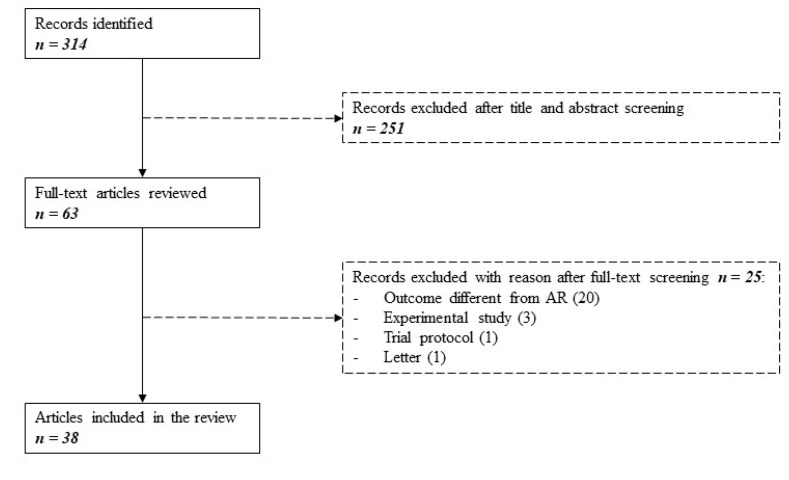
Search flowchart as per PRISMA guidelines. Three hundred fourteen studies were identified searching MEDLINE between January 2015 and May 2020. After evaluating for eligibility and inclusion, 38 articles were selected.

**Table 1 ijms-21-06889-t001:** Description of the study characteristics. The table summarizes the main characteristics of the 38 included studies.

Ref	Study Design	Single/Multicenter	Patients (*n*)	Enrolment (years)	Urinary Biomarker(s)	Ref. Standard	Outcome
Tinel [16]	Cross-sectional	Single center	329	2011–2016	CXCL9, CXCL10	Banff ‘15	TCMR, ABMR
Yang [17]	Cross-sectional	Multicenter	364	2010–2018	*Q score*	Banff ‘17	TCMR, ABMR
Kalantari [18]	Case-control	Single center	22	2016–2018	Unbiased metabolomics ^1^	Banff ‘97	TCMR
Verma [19]	Case-control	Single center	53	N/R	RNA-Seq signature	Banff ‘17	TCMR
Goerlich [20]	Case-control	Single center	39	2016–2017	T cells, TEC, PDX	Banff ‘13	TCMR, ABMR
Banas [21]	Cross-sectional	Single center	109	2011–2012	Unbiased metabolomics ^2^	Banff ‘09	TCMR, ABMR
Tajima [22]	Cross-sectional	Single center	80	2014–2016	LC3, CCL2, LFABP, NGAL, HE4	Banff ‘09	TCMR, ABMR
Kolling [23]	Case-control	Single center	93	N/R	Circular RNAs	Banff ‘09	TCMR
Sigdel [24]	Cross-sectional	Multicenter	150	2000–2016	*uCRM score*	Banff ‘09	TCMR, ABMR
Kim [25]	Case-control	Multicenter	23	N/R	Unbiased metabolomics ^3^	Banff ‘07	TCMR
Ciftci [26] *	Prospective	Single center	85	2014–2017	CXCL9, CXCL10	Banff ‘13	TCMR, ABMR
Banas [27]	Case-control	Single center	358	2008–2010; 2015–2016	Unbiased metabolomics ^2^	Banff ‘97	TCMR
Lim [28]	Case-control	Multicenter	47	2013–2015	Exosome proteins	Banff ‘07	TCMR
Chen [29]	Case-control	Single center	49	2006–2009	CXCL13	Banff ‘97	TCMR, ABMR
Barabadi [30] §	Cross-sectional	Single center	91	2013–2015	FOXP3	Banff ‘13	AR
Mockler [31] *^§^	Prospective	Single center	38	N/R	CCL2	Banff ‘13	TCMR
Ciftci [32] *	Prospective	Single center	65	2013–2015	TNFα	Banff ‘97	AR
Park [33]	Case-control	Single center	44	N/R	Exosome proteins	Banff (N/R)	TCMR
Millan [34] *	Prospective	Multicenter	80	N/R	miR-155-5p, CXCL10	Banff ‘97	TCMR
Seo [35]	Case-control	Multicenter	88	2013–2015	*CTOT4 formula*	Banff (N/R)	TCMR, ABMR
Gandolfini [36] §	Case-control	Multicenter	56	N/R	CXCL9	Banff ‘13	TCMR
Chen [37]	Case-control	Single center	156	2006–2009	sTim3	Banff ‘97	TCMR, ABMR
Domenico [38] §	Case-control	Single center	49	N/R	miRNA-142-3p	Banff ‘07	AR
Lee [39] §	Case-control	Single center	34	N/R	Donor-derived cfDNA	unclear	AR
Seeman [40] §	Case-control	Single center	15	2013–2014	NGAL	Banff ‘09	TCMR, ABMR
Blydt-H. [41]	Cross-sectional	Multicenter	59	2002–N/R	*ABMR score*	Banff ‘13	ABMR
Belmar V. [42] *	Retrospective	Single center	86	2012–2015	Albumin	Banff (N/R)	ABMR
Raza [43]	Cross-sectional	Single center	300	2009–2014	CCL2	Banff ‘97	TCMR
Galichon [44]	Cross-sectional	Multicenter	108	N/R	*CTOT4 formula*	Banff ‘09	TCMR, ABMR
Sigdel [45]	Cross-sectional	Single center	396	2000–2011	Unbiased proteomics	Banff ‘07	TCMR, ABMR
Garcìa-C. [46] ^§^	Cross-sectional	Single center	50	N/R	IL10, IFNγ	Banff ‘09	TCMR, ABMR
Ho [47] §	Cross-sectional	Single center	133	N/R	MMP7, CXCL10	Banff ‘07	TCMR
A. Elaziz [48]	Cross-sectional	Single center	54	2011–2014	PD1, FOXP3	Banff ‘07	TCMR
Lorenzen [49]	Cross-sectional	Single center	93	N/R	LncRNAs	Banff ‘09	TCMR
Rabant [50] *	Prospective	Single center	300	2010–2012	CXCL9, CXCL10	Banff ‘07	TCMR, ABMR
Rabant [51]	Cross-sectional	Single center	244	2011–2013	CXCL9, CXCL10	Banff ‘07	TCMR, ABMR
Blydt-H. [52]	Cross-sectional	Single center	51	2002–N/R	CXCL10	Banff ‘07	TCMR
Sigdel [53] §	Case-control	Single center	30	2000–2009	Exosome proteins	Banff ‘07	AR

Unbiased metabolomics: ^1^ (NAD, NADP, nicotinic acid, MNA, GABA, cholesterol sulfate, homocysteine); ^2^ (alanine, citrate, lactate, urea); ^3^ (guanidoacetic acid, methylimidazoleacetic acid, dopamine, 4-guanidinobutyric acid, and L-tryptophan). * Prediction study; § Diagnostic Test Accuracy analysis not present. N/R, not reported.

**Table 2 ijms-21-06889-t002:** Urinary biomarkers. Table illustrating all urinary biomarkers divided per category and in alphabetic order. Specific formulas and scores are also detailed.

Category	Biomarkers
**Cytokines**	
Chemokines Other	CCL2, CXCL9, CXCL10, CXCL13 IFNγ, IL10, TNFα
**Metabolites**	
Nucleotides Amino acids and Organic acids Other small molecules	NAD, NADP Alanine, Citrate, GABA, 4-Guanidinobutyric Acid, Guanidoacetic Acid, Homocysteine, Lactate, Methylimidazoleacetic Acid, Nicotinic Acid, l-Tryptophan Cholesterol Sulfate, Dopamine, MNA, Urea
**Proteins**	Albumin, LFAPB, HE4, LC3, MMP7, NGAL, sTIM3, Urinary extracellular vesicle (exosome) proteins (HPX, TSPAN1)
**RNAs** micro RNAs	Circular RNAs, FOXP3 mRNA, LncRNAs, PD1 mRNA, RNA-seq miR-142-3p, miR-155-5p
**Urinary Cells**	CD4+/CD8+ T cells, CD10+/EPCAM+ cells, PDX+ cells, TEC
**Scores and Formulas**	
ABMR score [41] CTOT-4 formula [54] Q score [17] uCRM score [24]	Signature of 133 unique metabolites CD3ε mRNA + CXCL10 mRNA + 18S rRNA Cell-free DNA + Clusterin + Creatinine + CXCL10 + Methylated Cell-free DNA + Total Urinary Protein 11 genes expression score on urinary cell pellet (including CXCL9 and CXCL10)

**Table 3 ijms-21-06889-t003:** QUADAS-2 tool assessment for DTA studies. Table illustrating risk of bias and applicability concerns evaluation as per QUADAS-2 tool for 29 studies providing diagnostic test accuracy data.

Ref	Risk of Bias				Applicability Concerns		
	Patient Selection	Index Test	Reference Standard	Flow and Timing	Patient Selection	Index Test	Reference Standard
Tinel [16]	☺	☺	☺	☺	☺	☺	☺
Yang [17]	☺	☺	☺	☺	☺	☺	☺
Kalantari [18]	☹	☹	☺	?	☹	☹	☺
Verma [19]	☹	☹	☺	☺	☹	☺	☺
Goerlich [20]	☹	☹	☺	☺	☹	☺	☺
Banas [21]	☺	☺	☺	?	☺	☺	☺
Tajima [22]	☺	☹	☺	☺	☹	☺	☺
Kolling [23]	☹	☹	☺	☺	☹	☺	☺
Sigdel [24]	☺	☹	☺	☺	☺	☺	☺
Kim [25]	☹	☹	☺	☺	☹	☺	☺
Ciftci [26] *	☹	☹	☺	?	☹	☺	☺
Banas [27]	☹	☹	☺	☺	☹	☺	☺
Lim [28]	☹	☹	☺	☺	☹	☺	☺
Chen [29]	☹	☹	?	☺	☹	☺	☺
Ciftci [31] *	☹	☹	?	?	☹	☺	☺
Park [33]	☹	☺	☺	☺	☹	☺	☺
Millan [34] *	?	☹	☺	?	☺	☺	☺
Seo [35]	☹	☹	☺	☹	☹	☺	☺
Chen [37]	☹	☹	?	☺	☹	☺	☺
Blydt-H. [41]	☺	☺	☺	☺	☺	☺	☺
Belm.V. [42] *	☹	☹	?	☺	☹	☺	☺
Raza [43]	☺	☹	☺	☹	☹	☺	☺
Galichon [44]	☺	☹	?	☺	☺	☺	☺
Sigdel [45]	☺	☺	☺	☺	☺	☺	☺
A. Elaziz [48]	?	☹	☺	☺	☺	☺	☺
Lorenzen [49]	☺	☹	☺	☺	☺	☺	☺
Rabant [50] *	☺	☹	☺	☺	☺	☺	☺
Rabant [51]	☺	☹	☺	☺	☺	☺	☺
Blydt-H. [52]	☺	☹	☺	☹	☺	☺	☺

*, Prediction study; ☺, Low Risk; ☹, High Risk; ?, Unclear Risk.

**Table 4 ijms-21-06889-t004:** Summary of the study results—Diagnostic studies with DTA. This table shows the outcome of diagnostic studies. Outcome and control group for the DTA analysis are reported (sample size when available), followed by test design, studied urinary biomarker(s), and thresholds when provided.

Ref.	Outcome (*n*)	Control Group (*n*)	Test Design, Biomarkers, Thresholds	Diagnostic Test Accuracy (95%CI)				
				Sens.	Spec.	PPV	NPV	AUC–Accuracy(%)
**Tinel** [16]	TCMR (17), ABMR (64), mixed (14)	ALL-B (normal, 21; IFTA, 154; BKVN, 23; ATN, 11; recurrent disease, 9; other, 78)	CXCL9 + CXCL10 for AR	62%	72%	41%	86%	**0.70 (0.64–0.76)**
			CXCL9 + CXCL10 for TCMR	79%	74%	21%	98%	**0.81 (0.73–0.89)**
			CXCL9 + CXCL10 for ABMR	72%	54%	28%	88%	**0.67 (0.61–0.74)**
**Yang** [17]	TCMR + ABMR (103)	ALL-B (normal, 170; bAR, 50;	Training: AR vs normal (Q score ≥ 32)	95%	100%	-	-	0.99 (0.99**–**1.00)
		BKVN, 9)	Validation 1: AR vs normal	91%	92%	-	-	**0.98 (0.96–1.00)**
			Validation 2: AR vs normal	100%	96%	-	-	**1.00 (1.00–1.00)**
			All AR vs All normal	95%	96%	87%	98%	**0.99(0.98–0.99)**
			All AR vs ALL-B	-	-	-	-	**0.96 (0.94–0.98)**
**Kalantari** [18]	TCMR (7)	DYS-B (normal, 15)	Unbiased metab.^1^	67**–**71%	40**–**100%	-	-	0.51**–**0.71
**Verma** [19]	TCMR (22)	ALL-B (normal, 28)	13-gene urinary cell signature	-	-	-	-	0.92 (0.85–0.99)
**Goerlich** [20]	TCMR (14) + ABMR (7)	DYS-B (normal, 18)	T cells + total TEC	-	-	-	-	0.90
			T cells + CD10+ TEC	-	-	-	-	0.89
			T cells + ECPAM+ TEC	-	-	-	-	0.91
			T cells + PDX+ cells	-	-	-	-	0.89
**Banas** [21]	TCMR + ABMR + mixed	ALL-B (normal) + STA	Unbiased metab.^2^	-	-	-	-	0.75 (0.68**–**0.83)
			Score = 3.0	91% (79**–**98)	34% (30**–**38)	-	-	-
			Score = 13.0	48% (33**–**63)	89% (86**–**91)	-	-	-
	+ bAR	+ (IFTA + other)		-	-	-	-	**0.71 (0.64–0.79)**
**Tajima** [22]	TCMR + ABMR (subclinical, 11)	STA-B (normal or borderline AR, 69)	LC3 (517.9 pg/mg)	64% (31–89)	78% (67–87)	32%	93%	0.73 (0.55**–**0.90)
			CCL2 (226.0 pg/mg)	82% (48–98)	57% (44–68)	23%	95%	0.69 (0.54–0.84)
			L-FABP (7.6 ng/mg)	9% (0–41)	88% (78–94)	15%	100%	0.61 (0.45–0.77)
			NGAL (12.8 ng/mg)	100% (72–100)	48% (36–60)	23%	100%	0.72 (0.59–0.84)
			HE4 (789.1 ng/mg)	100% (72–100)	54% (41–66)	26%	100%	0.81 (0.70–0.92)
**Kolling** [23]	TCMR (11; subclinical, 51)	STA-B (normal, 31)	hsa_circ_0001334 (2.41)	70% (59–80)	92% (64–100)	98%	32%	0.85 (*p* < 0.0001)
**Sigdel** [24]	TCMR + ABMR (45)	ALL-B (normal, 43; bAR, 19; BKVN, 43)	AR vs normal (uCRM score = 3.63)	95%	98%	-	-	0.99, *p* < 0.0001
			AR vs normal + bAR	87%	98%	-	-	-
			AR vs normal + bAR + BKVN	77%	98%	-	-	96.6%
**Kim** [25]	TCMR (14)	STA-B (normal, 17)	Unbiased metab.^3^	-	-	-	-	-
			Training: TCMR (10) vs STA-B (13)	90%	85%	-	-	0.93 (0.72–1.00) - 87%
			Validation: TCMR (4) vs STA-B (4)	-	-	-	-	**62.5%**
**Banas** [27]	TCMR	ALL-B (normal) + STA (extended)	Unbiased metab.^2^, train (180)	-	-	-	-	0.76 (0.69–0.82)
			Test (178) strict/extended cohort	-	-	-	-	0.72 (0.58–0.86)/ 0.74 (0.62–0.86)
**Lim** [28]	TCMR (25)	STA-B (normal, 22)	TSPAN1 + HPX	64%	73%	-	-	0.74
**Chen** [29]	TCMR (37) + ABMR (12)	ALL-B (normal, 58; CAN, 29; ATN, 10)	CXCL13 for AR vs. normal	84%	79%	-	-	0.82 (0.73–0.90)
			CXCL13 for AR vs. CAN + ATN	-	-	-	-	0.63 (0.52–0.75)
**Park** [33]	TCMR (22)	DYS-B (normal, 22)	iKEA					
			Training: TCMR (15) vs normal (15)	93%	88%	-	-	0.91 ± 0.02 - 90%
			Validation: TCMR (7) vs normal (7)	64%	100%	-	-	**0.84 ± 0.11 - 71%**
**Seo** [35]	TCMR (27) + ABMR (13)	STA-B (normal, 17); STA (22)	CTOT4 formula	-	-	-	-	0.72 (0.60–0.83)
			CXCL10 mRNA	-	-	-	-	0.72 (0.60–0.83)
			CD3ε mRNA	-	-	-	-	0.71 (0.60–0.83)
			18S rRNA	-	-	-	-	0.47 (0.33–0.60)
**Chen** [37]	TCMR (37) + ABMR (12)	STA-B (normal, 58)	sTim-3 (1.836 ng/mmol)	90%	83%	-	-	0.88 (0.81–0.95)
**Blydt-H.** [41]	ABMR (10)	ALL-B (normal, TCMR, transplant glomerulopathy, IFTA, other, 49)	ABMR score = 0.23	78%	83%	40%	96%	0.84 (0.77–0.91)
			ABMR score with top 10 metabolites	-	-	-	-	0.80 (0.73–0.88)
			Validation	-	-	-	-	**0.76 (0.67–0.84)**
**Raza** [43]	TCMR (acute, 101; borderline, 47; vascular, 17)	DYS-B (normal, 47; IFTA, 46) + STA (42)	CCL2 (198 pg/mL)	87%	62%	-	-	0.81 (0.76–0.86)
**Galichon** [44]	TCMR (11) + bAR (3) + ABMR (28) + mixed (9)	ALL-B (56)	CTOT4 formula	-	-	-	-	0.72 (0.61–0.82)
			CXCL10 mRNA	-	-	-	-	0.76 (0.66–0.86)
			CD3ε mRNA	-	-	-	-	0.67 (0.56–0.78)
			18S rRNA	-	-	-	-	0.63 (0.53–0.74)
**Sigdel** [45]	TCMR + ABMR (42)	ALL-B (normal, 47; CAN, 46; BKVN, 16)	Unbiased proteomics (11 peptides)					
			Validation: AR (20) vs normal (27), CAN (15), BKVN (16)	-	-	-	-	**0.94 (0.93–0.95)**
**A. Elaziz** [48]	TCMR (31)	STA-B (normal, 23)	PD1 mRNA (2.6)	80%	84%	-	-	0.81
			FOXP3 mRNA (1.5)	83%	90%	-	-	0.91
			PD1 + FOXP3 mRNA	94%	97%	-	-	0.98
**Lorenzen** [49]	TCMR (11; subclinical 51)	STA-B (normal, 31)	RNA L328 (9.556)	49%	96%	49%	93%	0.76 (*p* < 0.001)
**Rabant** [51]	TCMR (10) + ABMR (37) + mixed (31)	DYS-B (203)	CXCL9	58%	85%	59%	84%	0.71 (0.64–0.78)
			CXCL10	59%	83%	58%	84%	0.74 (0.68–0.80)
**Blydt-H.** [52]	TCMR (subclinical, 17; clinical, 9)	ALL-B (normal, 21; IFTA, 31)	CXCL10, subclinical (4.82 ng/mL)	59%	67%	-	-	0.81 (0.70–0.92)
			Clinical (4.72 ng/mL)	77%	60%	-	-	0.88 (0.73–1.0)

Results from a validation group are shown in **bold**. Unbiased metabolomics: ^1^ (NAD, NADP, nicotinic acid, MNA, GABA, cholesterol sulfate, homocysteine); ^2^ (alanine, citrate, lactate, urea); ^3^ (guanidoacetic acid, methylimidazoleacetic acid, dopamine, 4-guanidinobutyric acid, and L-tryptophan). ALL, all patients irrespectively of allograft function (-B, biopsied); DYS, dysfunctional graft patients (-B, biopsied); STA, stable graft patients (-B, biopsied).

**Table 5 ijms-21-06889-t005:** Summary of the study results—Predictive studies with DTA. This table shows the outcome of prediction studies. Outcome and control group for the DTA analysis are reported (sample size when available), followed by the studied urinary biomarker(s), thresholds and time from transplant to test.

Ref.	Outcome (n)	Control Group (*n*)	Biomarkers, Thresholds and Time Post-Transplant	Diagnostic Test Accuracy (95%CI)				
				Sens.	Spec.	PPV	NPV	AUC
**Ciftci** [26]	TCMR (9) + ABMR (6)	STA (70)	CXCL9, 1 day - 3 months	70–85%	37–88%	60–71%	71–90%	0.71–0.95
			CXCL10, 1 day - 3 months	78–82%	58–85%	59–73%	74–87%	0.75–0.97
**Ciftci** [32]	AR (9)	STA (56)	TNF-α (12.08 pg/mL), 1 day	71%	57%	-	-	0.74 (0.51–0.97)
			TNF-α (11.03), 7 days	100%	84%	-	-	0.95 (0.88–1.00)
			TNF-α (9.85), 1 month	100%	83%	-	-	0.91 (0.81–1.00)
			TNF-α (9.13), 3 months	100%	71%	-	-	0.83 (0.75–0.98)
			TNF-α (7.42), 6 months	100%	62%	-	-	0.82 (0.69–0.95)
**Millan** [34]	TCMR (8)	STA (72)	miR-155-5p (0.51), 1wk-6m	85%	86%	88%	100%	0.88 (0.78–0.97)
			CXCL10 (84.73 pg/mL),1wk-6m	84%	80%	90%	85%	0.87 (0.81–0.92)
			CXCL10:Cr (0.43), 1wk-6m	72%	73%	90%	96%	0.75 (0.67–0.83)
**Belm.V.** [42]	ABMR (subclinical)	ALL-B	Albuminuria (> 30 mg/g), 6m	-	-	-	-	0.75 (0.55–0.95)
**Rabant** [50]	AR (TCMR + ABMR + mixed, 76)	ALL-B	CXCL9:Cr (1.78 ng/mmoL),10d	61%	50%	24%	84%	0.58 (0.47–0.68)
			CXCL9:Cr (0.96), 1 month	81%	35%	23%	89%	0.50 (0.37–0.62)
			CXCL9:Cr (1.67), 3 months	57%	62%	18%	91%	0.57 (0.39–0.75)
			CXCL10:Cr (4.80), 10 days	57%	52%	23%	83%	0.54 (0.43–0.65)
			CXCL10:Cr (2.79), 1 month	83%	51%	29%	93%	0.72 (0.61–0.80)
			CXCL10:Cr (5.32), 3 months	54%	77%	25%	92%	0.68 (0.55–0.80)

ALL, all patients irrespectively of allograft function (-B, biopsied); STA, stable graft patients (-B, biopsied).

**Table 6 ijms-21-06889-t006:** Summary of the study results—Studies with no DTA. This table describes the main results from studies with no DTA. Sample size is reported for the outcome and control group when available.

Ref.	Outcome (n)	Control Group (*n*)	Biomarkers, Thresholds and Main Results
**Barabadi** [30]	AR (27)	ALL-B (normal, 45; CAN, 19)	FOXP3 mRNA expression was significantly higher in AR **(*p* < 0.001)**
**Mockler** [31] *	TCMR (5; borderline, 3)	STA-B	There was no significant association between 6 months post-transplant CCL2 and TCMR changes (*p* = 0.46)
**Gandolfini** [36]	TCMR (22)	ALL-B (normal, 19)	CXCL9 > 200 pg/mL in TCMR, 100-200 in dysfunction graft, and < 100 pg/mL in stable graft **(*p* < 0.01)**
**Domenico** [38]	AR (23)	ALL-B (ATN, 18; normal, 8)	mirRNA 142-3p was significantly higher in AR compared to stable graft **(*p* < 0.001)**; not compared to ATN **(*p* = 0.079)**
**Lee** [39]	AR (8)	STA (8); DYS-B (ATN, 8; other, 4)	Donor-derived cfDNA was not significantly different between groups (*p* = 0.95)
**Seeman** [40]	TCMR (2) + ABMR (2)	DYS-B (11)	NGAL was not significantly different between groups (*p* = 0.48)
**Garcìa-C.** [46]	AR (9)	ALL-B (fibrosis, 31; other, 10)	IL10 and IFNγ were not significantly different between groups (*p* = 0.95, *p* = 0.1)
**Ho** [47]	TCMR (17; subclinical, 17)	ALL-B (normal, 22)	MMP7 and CXCL10 were significantly elevated in subclinical **(*p* = 0.01, *p* < 0.0001)** and clinical **(*p* < 0.001)** TCMR
**Sigdel** [53]	AR (10)	DYS-B (IFTA, BKVN, 20)	Ten urinary exosomal proteins were significantly increased in AR **(*p* < 0.05)**

Statistically significant (*p* < 0.05) results are shown in **bold**. * Prediction study. ALL, all patients irrespectively of allograft function (-B, biopsied); DYS, dysfunctional graft patients (-B, biopsied); STA, stable graft patients (-B, biopsied).

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
