# Peer review of "Urinary Biomarkers for Diagnosis and Prediction of Acute Kidney Allograft Rejection: A Systematic Review"

_ijms, 2020, doi:10.3390/ijms21186889_

Round 1
Reviewer 1 Report
Since urine is easily collected and, thus, urinary biomarkers which are of use for diagnosis and prediction of AR after kidney transplantation. This systematic review paper is written well and analysis was based on the formal guidelines.
In recent 5 years, CXCL9 and CXCL10 seem to be the most promising biomarkers which can diagnose and predict the outcome of AR without graft's biopsy. From this paper, further studies using CXCL9 and CXCL10, or other biomarkers will be expected and confirmation of novel biomarkers of AR after kidney transplantation is expected.
Author Response
We really thank the reviewer for acknowledging the quality of our work.
Reviewer 2 Report
Introduction
"To our knowledge, the most recent systematic review 61 assessing urinary biomarkers’ ability for allograft AR diagnosis in kidney transplant patients 62 included papers published until 2015"
but there are two essential review articles:
1. Biomarkers of rejection in kidney transplantation
Singh, Neeraj Samant, Hrishikesh Hawxby, Alan Samaniego, Millie D
Current Opinion in Organ Transplantation: February 2019
2. Recent Advances on Biomarkers of Early and Late Kidney Graft Dysfunction
Marco Quaglia, Guido Merlotti , Gabriele Guglielmetti , Giuseppe Castellano , Vincenzo Cantaluppi
Int J Mol Sci . 2020 Jul 29;21(15):5404.
doi: 10.3390/ijms21155404.
The authors should have given short information concerning these articles.
Study Characteristics
„The majority of the included studies applied up to date Banff 97 classification, with ten studies apparently using the 1997 version or not reporting the year. Using an 98 outdated classification could be a source of bias mostly for studies assessing ABMR, as discussed 99 later. „
Very heterogenous studies in terms of considered outcomes. It would be worth to distinguish the studies in three groups concerning:
- TCMR, - ABMR – mixed AR
Moreover, it is important that the majority of cited articles were case control studies
Biomarkers
There is a very long list of markers in this chapter. This introduces disorder and difficulties in reading and understanding the content
This information can be showed by the type of marker (eg urinary chemokines, urine proteomics ..)in a table. Additionally used abbreviations CTOT4 formula, uCRM score and others should be explained.
Summary of the results
Table 3a-c showed the details of the role of the individual markers in the prediction and diagnosis of acute rejection in analysed studies. The informations contained in the table are repeated in the text of chapters 2.5.1, 2.5.2, 2.5.3. Reading these chapters was difficult for the reader. I propose to edit the chapters with their titles but add an overview of the results by marker type, e.g. chapter 2.5.2 T-cell mediated rejection diagnosis: add: 2.5.2.1 urinary chemokines, 2.5.2.2 mRNA
Discussion
I recommend to discuss the role of markers in diagnosis of the other pathology in renal transplants such as BKV infection, CNI toxicity, reccurent primary disease and compare the role in AR diagnosis in the subsections of the discussion.In addition, it is worthy to discuss the role of markers and the role of histological features of the graft biopsy including molecular microscope picture , separately.
Conclusions
I agree that the CXCL9 and CXCL10 chemokines are the most frequently used markers, as it was reported in the two review articles, which I mentioned above. Moreover, the attention should be paid to practical aspects for clinicians. Finally, the authors might have proposed a potential scheme for the use of these markers in clinical practice.
Round 2
Reviewer 2 Report
The manuscript was improved according to the comments. The new version is more understandable and legible.